

# Functional responses of a cosmopolitan invader demonstrate intraspecific variability in consumer-resource dynamics

Brett R. Howard[1], Daniel Barrios-O'Neill[2], Mhairi E. Alexander[3], Jaimie T.A. Dick[2], Thomas W. Therriault[4], Tamara B. Robinson[5] and Isabelle M. Côté[1]

[1] Earth to Ocean Research Group, Department of Biological Sciences, Simon Fraser University, Burnaby, British Columbia, Canada

[2] Institute for Global Food Security, School of Biological Sciences, The Queen's University Belfast, Belfast, Northern Ireland, United Kingdom

[3] Institute of Biomedical and Environmental Health Research (IBEHR), School of Science and Sport, University of the West of Scotland, Paisley, United Kingdom

[4] Pacific Biological Station, Fisheries & Oceans Canada, Nanaimo, British Columbia, Canada

[5] Centre for Invasion Biology, Department of Botany and Zoology, Stellenbosch University, Stellenbosch, Maiteland, South Africa

## ABSTRACT

**Background**. Variability in the ecological impacts of invasive species across their geographical ranges may decrease the accuracy of risk assessments. Comparative functional response analysis can be used to estimate invasive consumer-resource dynamics, explain impact variability, and thus potentially inform impact predictions. The European green crab (*Carcinus maenas*) has been introduced on multiple continents beyond its native range, although its ecological impacts appear to vary among populations and regions. Our aim was to test whether consumer-resource dynamics under standardized conditions are similarly variable across the current geographic distribution of green crab, and to identify correlated morphological features.

**Methods**. Crabs were collected from multiple populations within both native (Northern Ireland) and invasive regions (South Africa and Canada). Their functional responses to local mussels (*Mytilus* spp.) were tested. Attack rates and handling times were compared among green crab populations within each region, and among regions (Pacific Canada, Atlantic Canada, South Africa, and Northern Ireland). The effect of predator and prey morphology on prey consumption was investigated.

**Results**. Across regions, green crabs consumed prey according to a Type II (hyperbolic) functional response curve. Attack rates (i.e., the rate at which a predator finds and attacks prey), handling times and maximum feeding rates differed among regions. There was a trend toward higher attack rates in invasive than in native populations. Green crabs from Canada had lower handling times and thus higher maximum feeding rates than those from South Africa and Northern Ireland. Canadian and Northern Ireland crabs had significantly larger claws than South African crabs. Claw size was a more important predictor of the proportion of mussels killed than prey shell strength.

Corresponding author
Brett R. Howard, brett.howard@sfu.ca

**Discussion**. The differences in functional response between regions reflect observed impacts of green crabs in the wild. This suggests that an understanding of consumer–resource dynamics (e.g., the *per capita* measure of predation), derived from simple, standardized experiments, might yield useful predictions of invader impacts across geographical ranges.

## INTRODUCTION

The ever-increasing rate of introductions of species beyond their native ranges and the potential negative impacts on native biodiversity of species that become invasive continue to generate worldwide concern (*Seebens et al., 2017*). However, the effects of invaders are notoriously difficult to predict, especially across geographical ranges (*Simberloff et al., 2013*; *Doherty et al., 2016*). Many predatory invaders are responsible for large declines in the abundance and richness of native species (e.g., *Wiles et al., 1995*; *Medina et al., 2011*). These impacts are often attributed to advantages of invasive predators in novel environments, including the lack of prey resistance, release from natural enemies/pathogens, or behavioural, morphological, and physiological pre-adaptations (*Alpert, 2006*; *Sih et al., 2010*; *Weis, 2010*; *Roy et al., 2011*). However, not all introduced predators cause notable declines in native populations (*Gurevitch & Padilla, 2004*; *Zenni & Nuñez, 2013*); some have minimal detectable impacts on recipient ecosystems (*Simberloff & Gibbons, 2004*; *Hampton & Griffiths, 2007*; *Howard, Therriault & Côté, 2017*). These variable outcomes may arise because the impacts of an invasive predator are influenced by context-specific biotic and abiotic conditions (*Lipcius & Hines, 1986*; *Alcaraz, Bisazza & García-Berthou, 2008*; *Robinson, Smee & Trussell, 2011*; *Barrios-O'Neill et al., 2014*; *Paterson et al., 2015*). This variability can make it difficult to accurately predict the impacts of invasive species (*Dick et al., 2017*), especially when the same invasive species occurs at multiple locations (*Melbourne et al., 2007*; *Kumschick et al., 2015*).

Comparative functional response analysis (CFRA) has become a useful tool for elucidating relative variability in consumer–resource interactions among invasive species and under different contexts (*Barrios-O'Neill et al., 2014*; *Alexander et al., 2015*; *Paterson et al., 2015*; *Dick et al., 2017*). The functional response is the relationship between consumer (e.g., predator) consumption rate and resource (e.g., prey) density (*Holling, 1959*; *Holling, 1965*). This relationship provides information on the ability of a predator to find and consume prey and, by extension, its potential ecological impacts (*Dick et al., 2013*; *Dick et al., 2014*). Unlike predation studies, which seek to directly measure the impact of an invasive species in a particular location or on a particular species, the CFRA approach uses simplified experimental conditions to generate relative (not absolute) parameters that are comparable across contexts. Functional responses can be linear (Type I), hyperbolic (Type II), or sigmoidal (Type III) (*Holling, 1965*). The magnitude and type of functional response

can determine predator–prey coexistence (*Holling, 1959*; *Oaten & Murdoch, 1975*; *Hassell, 1978*). Type II responses in particular may potentially destabilize prey populations and lead to localized prey extinction (e.g., *Lipcius & Hines, 1986*; *Rindone & Eggleston, 2011*; *Spencer, Van Dyke & Thompson, 2016*). Studies using CFRA have consistently demonstrated that invasive species, ranging from plants (*Funk & Vitousek, 2007*) to invertebrates (*Dick et al., 2013*) and vertebrates (*Alexander et al., 2014*), consume available resources at a higher rate than analogous native species. While these results support the general concept that successful invasive species do well, in part, because they are more efficient at using resources, context-dependent biotic interactions or abiotic conditions can cause variation in invasive species functional responses (*Barrios-O'Neill et al., 2014*; *Barrios-O'Neill et al., 2016*; *Paterson et al., 2015*). It is thus unclear whether we should expect the functional responses of an invasive species to be conserved across geographical ranges or whether context differences between populations will result in variable functional responses. Intraspecific geographic comparisons of functional responses should make it possible to estimate the relative importance of local behavioural and morphological adaptations in determining invader responses to resource availability and their potential ecological impacts.

The European green crab (*Carcinus maenas*) is a well-known invasive species that occurs in intertidal and shallow subtidal habitats around the world (*Behrens Yamada, 2001*) (Fig. 1). Green crabs are viewed as highly effective generalist predators (*Gillespie et al., 2007*), with detrimental effects for native biodiversity in some regions (*Welch, 1968*; *Walton et al., 2002*; *Matheson et al., 2016*). However, there is large variation in the reported impacts among green crab populations, which does not simply relate to time since invasion. For example, on the east coast of North America, where green crab have been established since the 1800s (*Say, 1817*), significant declines in commercially important shellfish stocks have been attributed to green crab predation (*Glude, 1955*; *Welch, 1968*). There are also notable ecological impacts on shellfish species on the west coast of North America, where green crab have been established since the 1990s (*Grosholz et al., 2000*; *Grosholz et al., 2011*). In contrast, there are limited observed impacts by green crab populations in both Australia (introduced 1880s) and South Africa (introduced 1980s) (*Fulton & Grant, 1902*; *Le Roux, Branch & Joska, 1990*; *Carlton & Cohen, 2003*; *Hampton & Griffiths, 2007*); *Mabin et al., 2017*).

In this study, we investigate variability in consumer-resource dynamics of green crabs from regions within both their invasive and native ranges using CFRA. If green crab functional responses are variable among regions, we expect these differences to reflect local ecological impacts, as demonstrated in interspecific CFRA studies (e.g., *Dick et al., 2013*; *Alexander et al., 2014*; *Paterson et al., 2015*). Thus, crabs from populations within regions should have similar functional responses, but crabs from North American regions (in this study, Atlantic and Pacific Canada) might be expected to have higher functional responses than those from regions within the native range (in this study, Northern Ireland) and parts of the invaded range where their impacts appear limited (in this study, South Africa). We also investigated morphological differences among both crab and prey populations that might potentially cause inter-regional differences in functional responses.
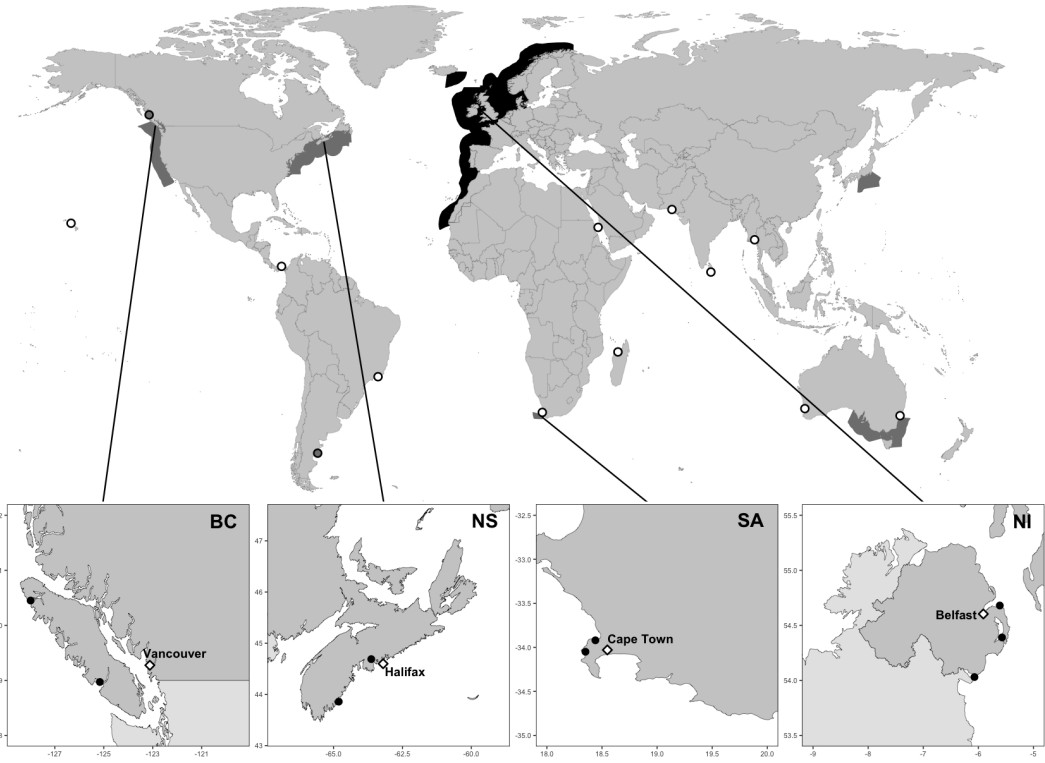

**Figure 1** **Global distribution of European green crab (*Carcinus maenas*) and sampling locations for green crab populations used in this study.** Native (black) and invaded (dark grey) ranges of European green crab (*Carcinus maenas*). Locations where green crabs occur in isolated populations are indicated by black dots. Open circles indicate locations where green crabs have been collected but established populations are not yet known to exist (see *Carlton & Cohen, 2003* for additional details). Insets show the sampling locations (black dots) for populations of green crabs used in this study: BC (British Columbia, Canada), NS (Nova Scotia, Canada), SA (South Africa), and NI (Northern Ireland, UK). White diamonds indicate locations of major cities near sampling locations.

## MATERIALS AND METHODS

### Site selection and animal collection

Green crabs were collected from nine populations from four regions where green crab have been introduced: British Columbia, Pacific Canada (BC, $n = 2$ populations), Nova Scotia, Atlantic Canada (NS, $n = 2$), and South Africa (SA, $n = 2$), and from the region where they are native: Northern Ireland, UK (NI, $n = 3$) (Fig. 1). A minimum of 18 crabs were collected from each site between July and September 2014 (Table S1). All crabs collected were males, with carapace widths between 55.0 and 65.0 mm, intact claws, and a firm shell (as springy or soft shells indicate recent moulting). Although the invasive congener *C. aestuarii* was previously recorded as co-occurring in very low densities alongside *C. maenas* in South Africa in the mid 1990s, they were no longer present a decade later (*Robinson et al., 2005*; *Robinson et al., 2016*). As such, all South African crabs were correctly identified as European green crab. Intertidal mussels of the genus *Mytilus* (BC: *M. trossulus*; NS: *M. edulis*; NI: *M. edulis*; SA: *M. galloprovincialis*) were used as prey because they are widely available in

all four regions (*Gosling, 1992*), are readily consumed by green crabs (*Elner, 1981*; *Morton & Harper, 2008*; *Behrens Yamada, Davidson & Fisher, 2010*), and are ecologically similar to one another (*Seed & Suchanek, 1992*). Mussels of 25 mm (±3 mm) length were collected by hand from a single site in each region, which standardized prey across populations within region. A similar standardization (i.e., using the same prey species) was not possible across regions, owing to ethical concerns about non-native species introductions.

Animals were housed in local research laboratories (BC: Bamfield Marine Sciences Centre; NS: Bedford Institute of Oceanography; NI: Queen's University Belfast; SA: Stellenbosch University). All crabs were housed in indoor tanks, with artificial lighting on day/night cycles similar to local summer conditions. In BC and NS, tanks were supplied with flow-through seawater from adjacent inlets. Tanks in SA and NI used artificial seawater systems. Across all trials, water temperatures varied across a narrow range (9–15 °C) suitable for green crab feeding (7–26 °C, *Behrens Yamada, 2001*). Salinity range (30–36‰) was also well within green crab tolerances (4–54‰; *Behrens Yamada, 2001*). Prior to and after being used in experiments, crabs in all locations were fed raw bait fish (e.g., herring) every two to three days. Prey animals were held separately from green crabs but under similar conditions.

## Experimental set-up and methods

At all locations, we used plastic bins (61 cm long × 40.6 cm wide), filled with seawater to a depth of 23 cm, as experimental chambers for all trials. The lids had a mesh screen to prevent escape while allowing light to diffuse inside the bins.

Prior to trials, green crabs were isolated and starved for 48 h to standardize hunger levels. Each crab was used only once. Intact mussels were cleaned of encrusting biota and checked for pre-existing damage. The evening prior to a trial, each bin received a randomly assigned prey density of two, four, eight, 16, 32, or 64 mussels, which were scattered across the bottom. The following morning, a single crab was placed into each bin and allowed to forage for eight daylight hours. Each prey density was replicated three times for each of the nine green crab populations tested. We retained, fed, and monitored all crabs for one week after testing to ensure that feeding behaviour had not been affected by imminent moulting. Because no moulting was observed, crabs that had eaten no prey (BC = 1/36 trials, NS = 2/35 trials, SA = 6/36 trials, NI = 15/54 trials) were retained in the analysis to reflect individual variation and because reduced consumption at low prey densities can be indicative of a Type III functional response. One Nova Scotia trial (at prey density = 2) was omitted owing to crab mortality. One predator-free control bin was run for every prey density and region to measure mussel mortality unrelated to predation.

## Morphological measurements

We evaluated morphological characteristics of both predator and prey that could cause differences in functional responses among populations. Claw size in green crabs is known to vary among populations (*Smith, 2004*; *Schaefer & Zimmer, 2013*), and claw strength is directly proportional to claw size, which has implications for handling times of crabs consuming shelled prey (*Behrens Yamada, Davidson & Fisher, 2010*). We therefore
measured crusher claw propal height as an index of claw size for each crab (*Behrens Yamada & Boulding, 1998*). Similarly, mussel shell thickness could influence consumption by green crab. We did not measure shell thickness of mussels in each trial, as mussels that were not consumed may have been rejected due to their thickness. Instead, in each region we collected an additional 19 to 30 randomly selected mussels of the same size as used in the trials, euthanized them and removed the tissue, keeping the valves intact. Shells were air-dried, measured and weighed to the nearest 0.01 g. Following *Freeman, Meszaros & Byers (2009)*, we calculated the shell thickness index (*STI*) as:

$$STI = Shell\ weight / [L * (H^2 + W^2)^{0.5} * \pi / 2]$$

where $L$, $H$, and $W$ correspond to linear measurements (in mm) of shell length (maximum anterior-posterior axis), height (maximum dorsal-ventral axis), and width (maximum lateral axis), respectively (*Lowen, Innes & Thompson, 2013*).

## Analysis

All analyses were done using R version 3.3.2 (*R Development Core Team, 2008*). Data were tested for homogeneity of variances and normality prior to statistical analyses to determine possible regional differences. The carapace width data were non-normal so a Kruskal–Wallis test was used, and the claw size data were heteroskedastic and thus a Welch's F test was used. We examined the relationship between the number of prey killed and average temperature (i.e., start temperature + end temperature/2) across all 64-mussel trials, using a linear mixed-effect model with region as a random effect. Temperature did not explain a significant amount of variation in number of mussels killed (Likelihood ratio test: $X^2 = 0.618$, $df = 1$, $P = 0.43$; Fig. S1). We therefore did not consider temperature in further analyses.

To determine functional responses as Type II or Type III, we first fit the proportion of prey consumed to prey density for each population using a logistic regression with the package 'frair' (frair::frair_test). Because the logistic regressions generated negative first-order terms in all cases, indicative of Type II functional responses (*Juliano, 2001*), we then fit the data using the appropriate random predator equation (see 'Results'), without prey replacement (*Rogers, 1972*):

$$N_e = N_0(1 - \exp(a(N_e h - T)))$$

where $N_e$ is the number of prey eaten, $N_0$ is the starting prey density, $a$ is the attack rate, $h$ is the handling time, and $T$ is the experimental duration. Values of $N_e$ and $N_0$ were generated experimentally, while $a$ and $h$ were estimated by fitting the model. Models were fit for each population using maximum likelihood estimation with the package function frair::frair_fit and bootstrapped ($n = 2,000$) to generate 95% confidence intervals.

Because functional responses were similar within regions (see 'Results', Figs. S2 and S3), we pooled populations within regions to test whether inter-regional differences were driven by differences in attack rate ($a$) or handling time ($h$). We re-fitted Rogers' Type II curves to regional data and bootstrapped the fits ($n = 2,000$) to generate parameter estimates for $a$, $h$ and maximum feeding rates ($1/hT$). The 95% confidence intervals for these parameter estimates were first compared visually and then more formally where necessary (Table S2).

Finally, to identify factors underpinning regional differences in prey consumption, we used generalized linear mixed-effects models (GLMMs) with a binomial error distribution to predict the proportion of prey consumed by green crabs as a function of claw size, prey STI, and region using a suite of additive candidate models. Mean prey STI for each region was included as a continuous, fixed effect. Because attack rates and handling times by green crabs from both Canadian regions (see 'Results') were similar, we combined BC and NS into a single region (North America, NA) for comparison with SA and NI. Finally, we included initial prey density as a fixed effect—not as an explanatory variable *per se* but because it is important in functional responses—and population as a random effect in all candidate models. The best-supported model was identified using Akaike's Information Criterion corrected for small sample sizes (AICc), where the top model had the lowest AICc value (*Burnham & Anderson, 2002*). We also determined the relative variable importance (RVI) of each fixed effect, based on the sum of the AICc weights for models that included the focal variable (*Burnham & Anderson, 2002*), and the marginal and conditional $R^2$ values for the top model (*Nakagawa & Schielzeth, 2013*). To display the individual effect of each variable included in the top model on the predicted proportion of mussels killed, we used the 'effects' package to calculate effect sizes for each variable, relative to the mean values (continuous data) or proportional distribution (categorical data) of the other factors in the model (*Fox, 2003*; *Fox & Hong, 2009*).

# RESULTS

## Regional patterns of functional responses

In trials without crabs, 100% of mussels survived. All logistic regressions indicated the predation data were best fit using Type II functional response models. Within regions, the confidence intervals around the number of prey killed overlapped between populations at most prey densities, indicating that differences in predatory behaviour within regions were minimal (Figs. S2 and S3). Inter-regionally, we found the highest functional response curves for North American green crab (BC and NS) (Fig. 2). Attack rates (*a*) were highest in BC, NS, and SA and the lowest in NI, but there was overlap in confidence intervals between all regions except BC and NI and NS and NI (Fig. 3A, Table S2). Handling times (*h*) were lower in BC and NS than in SA and NI, with no overlap of confidence intervals between these two groups (Fig. 3B, Table S2). By extension, the maximum feeding rates of North American green crabs were considerably higher than those of crabs in SA or NI (Fig. 3C).

## Potential drivers of regional variation in prey consumption

Although there was no significant difference in crab carapace width among regions (Kruskal–Wallis test, $P = 0.68$), claw size did differ significantly among regions (Welch's $F_{3,84.55} = 40.28$, $P < 0.01$), with crabs from NI, BC and NS having the largest claws and those from SA, the smallest (Fig. 4A). Mussel shell thickness index (STI) also differed significantly among regions (Kruskal–Wallis test, $P < 0.01$), resulting in a clear regional ranking (SA>NI>BC>NS) of decreasing mussel shell thickness (Fig. 4B).

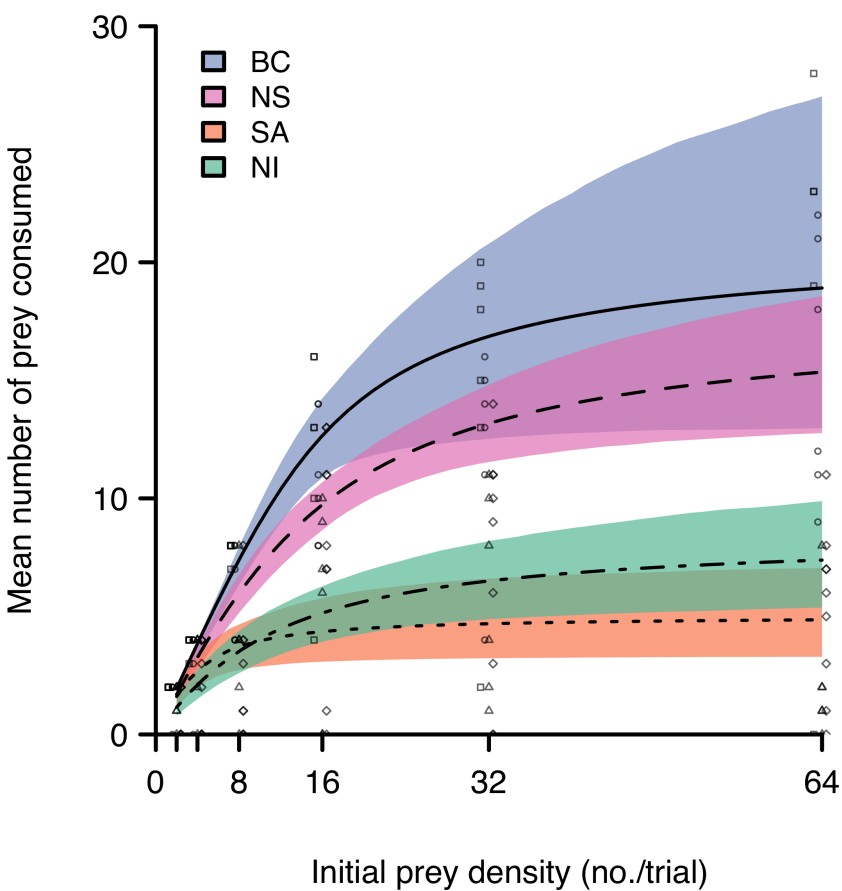

**Figure 2  Functional response curves for European green crab preying on mussels (*Mytilus* spp.) in four regions.** Functional response curves, modeled from the raw data (open symbols) with a Type II Rogers random predator equation without prey replacement, for European green crab preying on mussels (*Mytilus* spp.) in four regions: BC (British Columbia, Pacific Canada; solid line; open square), NS (Nova Scotia, Atlantic Canada; dashed line; open circle), SA (South Africa; dotted line; open triangle), and NI (Northern Ireland, UK; dot-dashed line; open diamond). The mean number of prey consumed by green crab in each region has been averaged across the multiple populations shown in Fig. S3. Shaded areas represent the 95% bootstrapped confidence intervals.

Prey density was the most important variable (RVI = 1.0) and was included in all models of proportion of prey consumed by green crabs. Region (RVI = 0.98), claw size (RVI = 0.85) were the next most important variables across all models. Prey STI (RVI = 0.48) was relatively less important.

There was substantial support for two of the candidate models (Table 1). Both included claw size and region as important predictors of the proportion of mussels killed. The second-ranked model also included prey STI, but this variable had poor explanatory power: it did not substantially improve the model fit (as indicated by the log-likelihood) or the marginal $R^2$ (Table 1).

The variables in the top model were prey density, claw size, and region (Table 1). Increasing prey density resulted in proportionally fewer mussels being killed, as expected

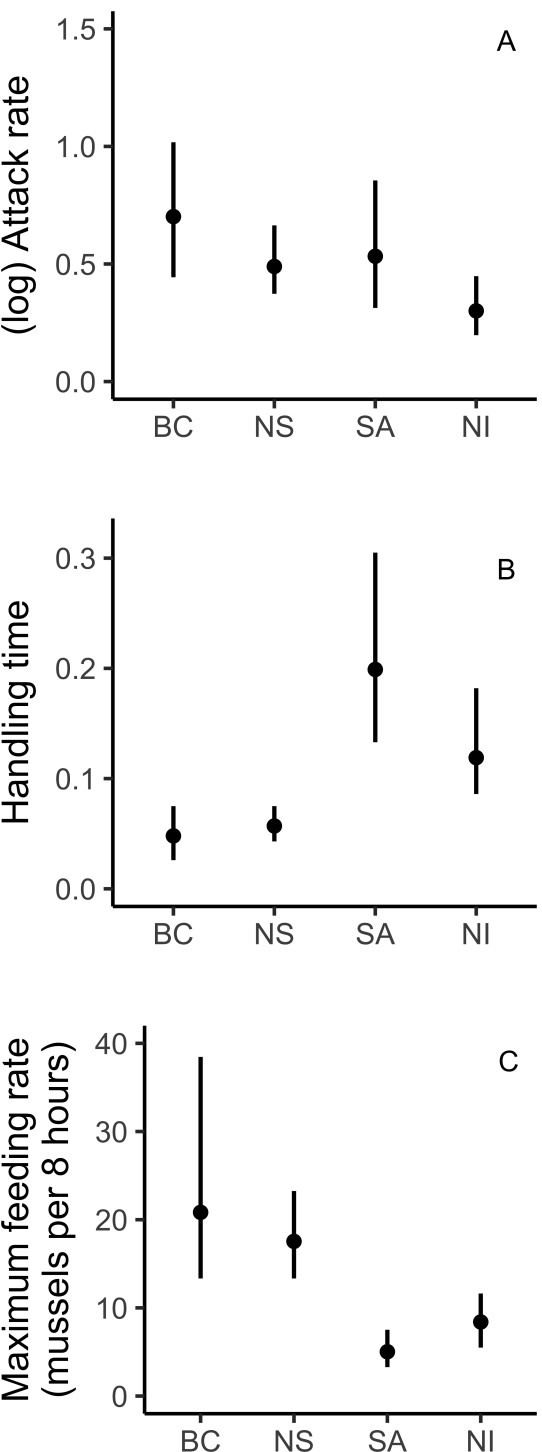

**Figure 3  Parameter estimates of attack rate *a*, handling time *h*, and maximum feeding rate 1/*h* T for European green crabs feeding on mussels.** Parameter estimates (±95% CI) of (A) attack rate *a*, (B) handling time *h*, and (C) maximum feeding rate 1/*h* T, from bootstrapped Type II functional response curves of green crabs preying on varying densities of mussels. Green crabs were collected from BC (British Columbia, Pacific Canada), NS (Nova Scotia, Atlantic Canada), SA (South Africa), and NI (Northern Ireland, UK).

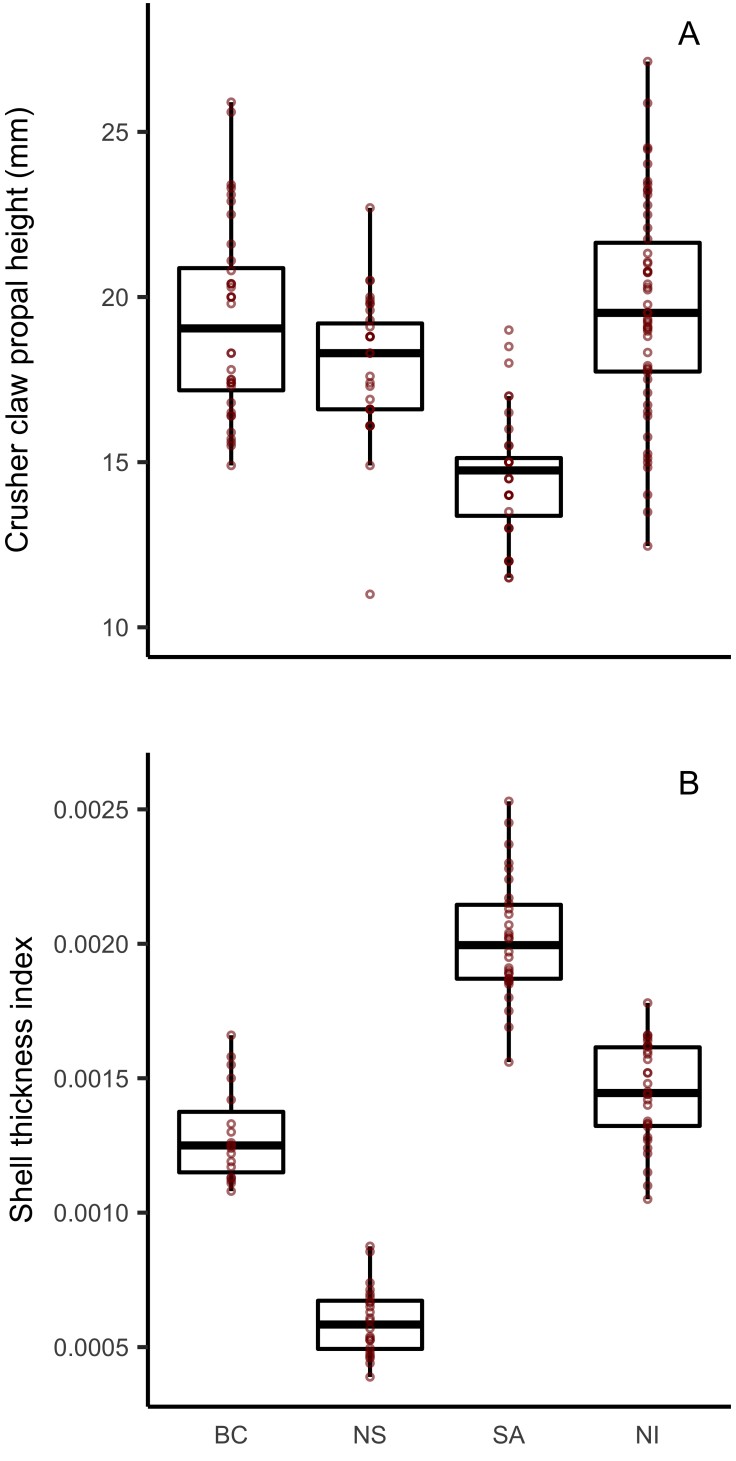

**Figure 4** **Regional variation in European green crab claw size and *Mytilus* mussel shell thickness for four regions.** Regional variation in potential determinants of proportion of mussels killed by European green crabs in four regions: BC (British Columbia, Pacific Canada), NS (Nova Scotia, Atlantic Canada), SA (South Africa), and NI (Northern Ireland, UK). (A) Claw size (i.e., propal height, in mm) of crabs, and (B) mussel shell thickness index. Raw data indicated by open circles.

**Table 1  Results of model selection on all candidate binomial generalized linear mixed-effects models of variation in the proportion of mussels killed by European green crabs in functional response trials.** Results of model selection using Akaike's Information Criterion AICc, showing all candidate binomial generalized linear mixed-effects models of variation in the proportion of mussels killed by European green crabs in functional response trials. Fixed effects included crab claw size, region (North America, South Africa, or Northern Ireland), and the shell thickness index (STI) of mussels from each region. Prey density per trial (density) was included as a fixed effect, and crab population as a random effect in all models. $k$ is the number of parameters in each model; $\Delta AIC_c$ is the difference in AICc value between the focal model and the model with the lowest $AIC_c$; Akaike weight $w_i$ is interpreted as the probability that a given model is the best model of the candidate set given the data at hand. Marginal $R^2$ values are also given as an index of model fit.

| Model | $k$ | LL | $AIC_c$ | $\Delta AIC_c$ | $w_i$ | Cumulative $w_i$ | Marginal $R^2$ |
|---|---|---|---|---|---|---|---|
| Density + claw + region | 6 | −474.5 | 961.58 | 0 | 0.44 | 0.44 | 0.29 |
| Density + claw + STI + region | 7 | −473.5 | 961.79 | 0.21 | 0.40 | 0.84 | 0.29 |
| Density + STI + region | 6 | −476.3 | 965.17 | 3.59 | 0.07 | 0.91 | 0.29 |
| Density + region | 5 | −477.4 | 965.17 | 3.59 | 0.07 | 0.98 | 0.28 |
| Density + claw + STI | 5 | −479.5 | 969.47 | 7.89 | 0.01 | 0.99 | 0.23 |
| Density + claw | 4 | −481.3 | 970.83 | 9.26 | 0 | 1.00 | 0.18 |
| Density + STI | 4 | −482.2 | 972.58 | 11.00 | 0 | 1.00 | 0.22 |
| Density-only | 3 | −484.2 | 974.55 | 12.97 | 0 | 1.00 | 0.17 |
| Intercept-only | 2 | −733.88 | 1,471.84 | 510.26 | 0 | 1.00 | 0 |

from Type II functional responses that reach saturation (Fig. 5A). The proportion of mussels killed increased with claw size, as expected, regardless of region and prey density (Fig. 5B). At mean prey density and claw size, mussels had a 33% chance of being killed by green crabs in NI and SA (Fig. 5C). In contrast, and as predicted, green crabs in North America imposed the highest prey mortality. The probability of a mussel being killed in North America was 67% (Fig. 5C). Together, these three fixed effects in the top model explained 29% of variation in the proportion of prey killed (marginal $R^2$). The combination of the fixed effects and random effect (population) explained 31% of this variation (conditional $R^2$). To validate our approach of combining the North American populations we also re-ran the analysis on a modified version of our top model that included all four regions, rather than three, to see if there were any differences in the results. All the trends were consistent with our top model (Fig. S4), and there were no differences in the marginal and conditional $R^2$ values or the model coefficients for prey density and claw size.

## DISCUSSION

Comparative functional response analysis (CFRA) has consistently demonstrated that the functional responses of invaders reflect their known ecological impacts, but it has traditionally focussed on interspecific comparisons between invasive and native species (e.g., *Dick et al., 2013*; *Alexander et al., 2014*; *Paterson et al., 2015*). CFRA has not previously been used to investigate geographic variation in functional responses of a single, cosmopolitan invader. Here, we did not observe large differences in the functional responses of green crabs from populations within regions: Although it cannot be assumed our results apply to entire ranges, as this would require more extensive sampling in both North America and Europe, crabs from populations several hundred kilometres apart but in the same region showed similar attack rates, prey handling times, and maximum feeding

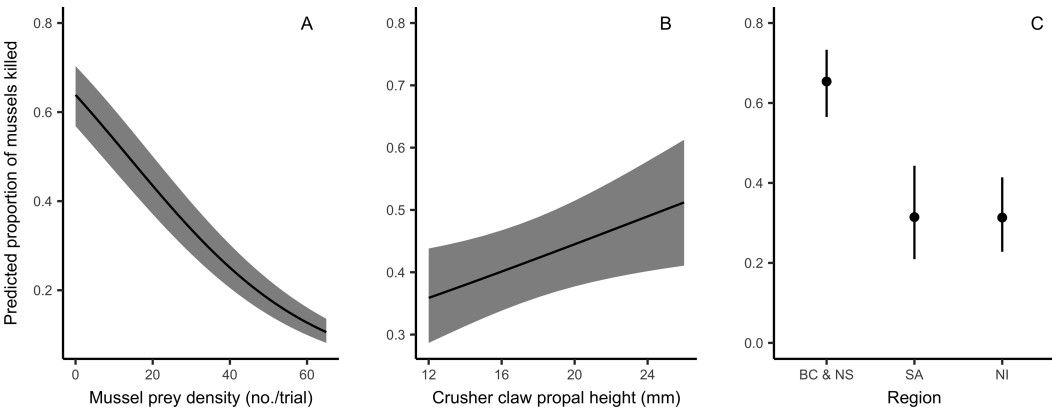

**Figure 5** **Predicted proportion of mussels killed by European green crabs in relation to prey density, claw size (mm), and the region of origin, derived from top generalized linear mixed-effects model.** The predicted proportion of mussels killed by European green crabs (with 95% confidence intervals) in relation to (A) prey density, (B) claw size (mm), and (C) the region from which crabs were collected: North America (British Columbia and Nova Scotia, BC & NS), South Africa (SA), and Northern Ireland, UK (NI). Predictions are derived from a generalized linear mixed-effects model (see top model in Table 1), and are shown for each factor when the other factors are fixed at their mean or proportional values.

rates. However, differences in functional response curves and parameters among regions were large. Furthermore, the higher functional responses of invasive green crabs from North America compared to South Africa and Northern Ireland appear to reflect, at least qualitatively, their predatory impacts in the wild (see *Glude, 1955*; *Welch, 1968*; *Grosholz et al., 2000*; *Grosholz et al., 2011*; *Mabin et al., 2017*). Despite their experimental simplicity, functional responses might therefore be a useful, relative estimate of predation that can help inform predictions about the ecological impacts of green crabs in areas where more refined impact predictions are not yet available or impossible to make.

We found regional differences in green crab capacity to kill mussels. These differences stemmed mainly from differences in handling times, and by extension maximum feeding rates (which are mathematically derived from handling times), although some variation in attack rates was also observed. The foraging success of predators depends on their behaviour, morphology, and physiology as they detect, attack, capture, and consume prey (*Hassell, 1978*; *Lima, 2002*). Attack rate ($a$) reflects the first two steps of this foraging sequence. We found some inter-regional differences in attack rates, with invasive crabs from BC demonstrating higher attack rates than those in Northern Ireland. Handling time ($h$), by comparison, reflects the time it takes for a predator to capture and consume prey items and is influenced by physiological and morphological constraints on the predator (*Elner & Hughes, 1978*; *Hassell, 1978*; *Vucic-Pestic et al., 2010*). Handling times were markedly lower, and maximum feeding rates ($1/h$ T) higher, for green crabs from North America than for those from South Africa and Northern Ireland. Interestingly, higher resource consumption by invasive species, compared to native species, is usually realized either through higher attack rates (e.g., *Dick et al., 2013*) or lower prey handling times (e.g., *Bollache et al., 2008*; *Haddaway et al., 2012*; *Alexander et al., 2014*), but rarely both. Differences in attack rates

among regions might have been driven by differences in individual activity levels, which are often higher in invasive than in native species (*Sih, Bell & Johnson, 2004*). In contrast, differences in claw size likely drove the observed differences in handling times. Green crabs from North America (BC and NS) had significantly larger claws, and shorter handling times, than crabs from South Africa. Crabs with larger claws have a morphological advantage over crabs with smaller claws, because large claws reduce the effort required to break mussels and the risk of claw damage (*Behrens Yamada, Davidson & Fisher, 2010*). This suggests that invasive green crabs from North America are morphologically better suited to handling hard-shelled prey than those from South Africa. Contrary to the pattern, however, native green crabs in Northern Ireland had large claws, on par with those of North American green crabs, but their handling times were significantly lower, more closely matching those seen in South African crabs. Because handling time incorporates both breaking time and eating time (*Elner & Hughes, 1978*; *Lee & Seed, 1992*; *Smallegange & Van der Meer, 2003*; *Calderwood, O'Connor & Roberts, 2016*), perhaps crabs in Northern Ireland are under less pressure to 'eat quickly' due to less competition or kleptoparasitism (*Smallegange, Van der Meer & Kurvers, 2006*; *Chakravarti & Cotton, 2014*), while still requiring large claws to crush thick-shelled local mussels.

There are four possible explanations for inter-regional differences in claw size and prey handling times. First, differences could be primarily driven by genetic variation. While there are detectable founder effects in some green crab populations (*Darling et al., 2008*), genetic variation does not explain the large phenotypic variation seen, including in claw size, within the native range of green crabs (*Brian et al., 2006*). It therefore seems unlikely that inter-regional variation in claw size is linked to a variable genetic make-up of founder individuals. Second, claw size could be a highly plastic trait. Green crabs can modify their claw sizes in response to prey shell thickness (*Brian et al., 2006*; *Schaefer & Zimmer, 2013*). This phenotypic response occurs under laboratory conditions (*Baldridge & Smith, 2008*) and along biogeographic gradients (*Smith, 2004*). In our study, claw size did not covary with prey shell thickness. However, the standardized mussel prey we offered might have not always reflected local diets of green crabs. For example, green crabs in British Columbia are currently only found in soft-sediment habitats where their diet consists mainly of infaunal clams (*Klassen & Locke, 2007*), which can have very thick shells (*Boulding, 1984*). In contrast South African green crabs eat predominantly small gastropods and soft-bodied prey (e.g., polychaetes) (*Le Roux, Branch & Joska, 1990*). Claw size may therefore normally be more closely linked to prey characteristics than our results suggest. Third, differences in water temperatures could affect the calcification of crab exoskeletons and of their molluscan prey. Warmer temperatures lead to decreased calcification, so crabs in warmer habitats may therefore have weaker claws with which to attack shelled prey. However, because the effect of decreased calcification would also make prey shells weaker, handling times should be unaffected overall (*Landes & Zimmer, 2012*). Finally, inter-regional variation in claw sizes, and by extension handling times, may reflect selective forces beyond prey defenses, including reproduction (mate competition) and agonistic interactions (interference competition) (*Lee & Seed, 1992*). Claw size is the best determinant of success in intra- and interspecific agonistic interactions between crabs

(*Lee & Seed, 1992*; *Sneddon, Huntingford & Taylor, 1997*). It is notable that green crabs in North America face competition from large-clawed decapods like Dungeness crab (*Metacarcinus magister*) in BC and American lobster (*Homarus americanus*) in NS (*McDonald, Jensen & Armstrong, 2001*; *Rossong et al., 2006*). A combination of exposure to thick-shelled prey and a highly competitive environment could explain the especially large claws and fast handling times of green crabs sampled from North America compared to those from other populations.

Globally, the impacts of green crabs seem to vary among regions, with some populations (e.g., North America and Tasmania) appearing to have larger impacts on intertidal communities than others (e.g., South Africa, Australia, or Japan) (*Behrens Yamada, 2001*). The quantitative evidence for this variability is provided by a few large-scale field studies showing that green crabs are effective bivalve predators that have negatively impacted native community composition, trophic interactions, critical habitat, and human economic interests (*Welch, 1968*; *Grosholz et al., 2000*; *Walton et al., 2002*; *DeRivera, Grosholz & Ruiz, 2011*; *Matheson et al., 2016*). Our finding that green crabs sampled from North America have higher attack rates and lower prey handling times than those sampled from other regions is consistent with these field observations. Moreover, North American green crabs have had markedly different patterns of spread than in other regions, including in South Africa where green crabs have a comparatively restricted range despite becoming established decades ago (*Mabin et al., 2017*). The limited success and impacts of invasive green crabs in South Africa have been attributed to abiotic conditions (e.g., fast-flowing water and highly exposed coasts) being unfavourable to range expansion (*Le Roux, Branch & Joska, 1990*; *Robinson et al., 2005*; *Hampton & Griffiths, 2007*). Our results demonstrate that South African green crabs exploit a similar prey less effectively than green crabs from other invasive regions. This suggests that the variable success of different populations of green crabs is partially driven by biotic interactions, not just habitat suitability.

The CFRA approach has been successful because it entails an extreme reduction of the complexity of experimental conditions. Functional response studies do not seek to generate absolute values of foraging parameters under realistic environmental and other contexts (e.g., habitat structure). Instead, the approach generates relative parameters that are comparable across species and contexts. Thus, high-impact invasive species typically display functional response curves that are steeper and/or have higher asymptotes than similar native species or lower-impact invaders (e.g., *Dick et al., 2013*; *Alexander et al., 2014*; *Paterson et al., 2015*). Our study is the first to establish that there is also marked inter-regional variation in the functional responses of a globally invasive consumer that appears to reflect, at least qualitatively, the regionally variable impacts of green crabs.

## CONCLUSIONS

CFRA can be a powerful approach with which to compare the relative impacts of invasive consumers both within and among species. As it relates to European green crab, it would be interesting to apply the method used here to populations of green crab we were not able to cover, such as those in Australia, the more southern parts of the North American ranges,

and elsewhere in the native range. This method could also be applied to native decapod species that co-occur with green crab to help identify how much competition influences foraging behaviours. Finally, while we make the inference that the functional responses described here may reflect impacts of green crabs in the field, data gaps in the literature make it difficult to be more definitive about that relationship or use these functional response results predictively. Methods to link experimental functional responses to field impacts exist (e.g., *Parker et al., 1999*; *Dick et al., 2017*), but at a minimum require data on abundance that is largely unavailable for European green crab. Where this information is available, functional responses offer a simplified, standardized metric of *per capita* impact that can be used to predict the ecological impacts of invasive species.

## ACKNOWLEDGEMENTS

We thank Anthony Ricciardi and Josephine Iacarella for their input in the development of the question and approach. BRH thanks Claudio DiBacco and Jennifer Fitzgerald for their hospitality, and Sarah Calbick, Nate Feindel, Chris McCarthy and Sean Godwin for their assistance.

### Funding

Funding was provided by a scholarship from the Second Canadian Aquatic Invasive Species Network (CAISN II) to Brett R. Howard and a Natural Sciences and Engineering Research Council of Canada (NSERC) Discovery Grant to Isabelle M Côté. The funders had no role in study design, data collection and analysis, decision to publish, or preparation of the manuscript.

### Grant Disclosures

The following grant information was disclosed by the authors:
Second Canadian Aquatic Invasive Species Network (CAISN II).
Natural Sciences and Engineering Research Council of Canada (NSERC).

### Competing Interests

The authors declare there are no competing interests.

### Author Contributions

- Brett R. Howard conceived and designed the experiments, performed the experiments, analyzed the data, contributed reagents/materials/analysis tools, prepared figures and/or tables, authored or reviewed drafts of the paper, approved the final draft, led the writing of the manuscript.
- Daniel Barrios-O'Neill and Mhairi E. Alexander conceived and designed the experiments, performed the experiments, analyzed the data, contributed reagents/materials/analysis tools, authored or reviewed drafts of the paper, approved the final draft.
- Jaimie T.A. Dick, Thomas W. Therriault and Tamara B. Robinson conceived and designed the experiments, contributed reagents/materials/analysis tools, authored or reviewed drafts of the paper, approved the final draft.
- Isabelle M. Côté conceived and designed the experiments, contributed reagents/materials/analysis tools, authored or reviewed drafts of the paper, approved the final draft, led the writing of the manuscript.

## Data Availability

Raw data can be found in the Supplemental Information.

## Supplemental Information

Supplemental information for this article can be found online at http://dx.doi.org/10.7717/peerj.5634#supplemental-information.

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
