# Peer review of "Functional responses of a cosmopolitan invader demonstrate intraspecific variability in consumer-resource dynamics"

_PeerJ, doi:10.7717/peerj.5634_

## Round 0.1 · original submission · Major Revisions

First of all, apologies for taking so long to come back to you. We have been waiting for a third review than eventually did not come, so my decision is based on the 2 reviews we have in hand. Although both reviewers agree that your work adresses an interesting topic, they also express their concern about the appropriateness of the experimental design used and specifically on the number of replicates. You will see that they differ in the extent of the changes necessary to make your manuscript acceptable for publication. Based on their reports I consider that you need to carry out major revisions.

Reviewer #2 has serious concerns about different aspects of the experimental design you used, ranging from the choice of the 4 regions to the actual way the feeding experiments were performed, with single food choices and handling times in need of clarification. Reviewer #1 also expresses his/her doubts about the number of replicates and the taxonomic identification of two closely related species (C. maenas and C. aestuarii), at least coexisting in South African samples. Altogether, their concerns potentially affect the validity of your findings.

Additionally, both reviewers have identified several technical and editorial issues that need to be carefully considered in the revision. Please prepare a revised version of your manuscript detailing how you have addressed each reviewers’ comment on a point-per-point basis. You should pay specific attention to the figures content and format as noted by Reviewer #2 Depending on the way you have addressed this reviewer’s comments I may have him/her look at it again.

We are looking forward to receiving your revised manuscript.

Reviewer 1 ·

Basic reporting

This is a well-written piece of research that aims at comparing the role of different populations of an invasive crab species in several areas, both native and colonised, concerning predator-prey functional responses. There are however some points, noted below, that need clarification.

The authors have a good background and the references used are relevant. There are however some points, that need to be further developed.

The structure of the article is correct. The section on Results is however rather short when compared with the rest of the sections.

SPECIFIC COMMENTS
Abstract
- Line 28: the acronym is not necessary here.
- L 30 I would change the sentence by something like: The European Green Crab Carcinus maenas is a northeast Atlantic intertidal and shallow water species which has become invasive in several regions around the world (it is not a globally invasive predator "per se". It's not its fault, but probably ours).
- L40: "Type II functional response". Not adequate in the Abstract. This is specialist jargon; not many readers may know what this means (myself, for example). It should be explained in a more intelligible way.
- L42: A more precise meaning of "higher attack rates" should be presented here. By itself, I could not visualize what an attack rate could be.
-
Introduction
- First sentence, L59: As indicated above, species are not invasive per se. Some of them have become invasive in some areas.

Experimental design

The experimental design is simple and robust, but it probably would need a larger number of replicates.

Both C. maenas and C. aestuarii are present in South African waters. This should have been taken into account and could modify conclusions reached. No reference at this "problem" is found in the manuscript.
- The sample size used (a minimum of 18 male crabs) by study area is not that large. A slightly larger sample size should not be that difficult to obtain for Carcinus. Sample sizes larger than 25-30 individuals have been shown to strongly diminish overall variability in means and proportions in many population biology studies. I recognize the effort done to sample and run the experiments, but this point should be clarified.
- Given the importance of handling time and of the success rate when running the experiments, information on moult stage of the crabs used should be given. Information on "red" vs "green" morphotypes, which is a proxy of the time spent since the last moult could perhaps be obtained from pictures of the individual crabs used, if they are available.
- L227: I have some doubts on the grouping /combining of British Columbia (Pacific) and Nova Scotia (Atlantic) into a single group for further comparisons (even though results are not significantly different). The study areas are so different and separated that I think this is not necessary.

Validity of the findings

There is one main problem concerning the experimental design. I assume that the authors have identified correctly the crab species they are dealing with. There are two species in the genus Carcinus: C. maenas, and C. aestuarii, sometimes somewhat difficult to morphologically identify, and can be identified through genetics analyses. Both species, especially C. maenas, have shown invasive characteristics and are, or have been present in at least one of the study areas, South Africa. There is no reference to this fact in this manuscript. Some of the differences found may be related to the use of, at least some, crabs belonging to C. aestuarii.

Discussion

- Calcification differences with temperature,both in predator and prey, as well as co-evolution between predator and prey, are points worth of being discussed.

Additional comments

No further comments

Reviewer 2 ·

Basic reporting

Some of the figures are difficult to read. For example, for Figure 1 native and invaded ranfes are difficult to identify due to the selected colours (black and dark gray).
Same proble for figues 2 and 3. No clear and "clean" distinction beteween blacka and darke grey.
Figures 2 and 3 offer redundant information. Keep only one of them. I suggest to keep Figure 3.
A more professional aspect for figure 5 should be great

All other aspects are fine for me.

Experimental design

The comparison between invaded and native range is flawed. Three different regions are used for invaded but only one for native range (Ireland). No formal comparison is possible. The native region is un-replicated and this lead to confounding (or pseudoreplication). With the described design, the interpretation should be limited to compare differences among 4 regions, but formal discussion van be done on the differences between native and invaded range.
In general, I don’t not like feeding experiment made in the laboratory. At best, they have difficult interpretations. In most cases, animals are forced to consume the preys that may be unfamiliar for the consumer. And it is done under environmental conditions far from natural conditions. Extrapolation to the field may be flawed. Of course this is a very biased point of view.
Starvation is not a very good idea in feeding experiments, as it can artificially influence the response of the consumers
I also have serious doubts on the validity of single choice experiments as it may force unnatural behaviours in the consumers. This is relevant for the exact shape of the functional response. Assuming that a handling time exist, only a type II response should be expected. With a single food item, Type III responses cannot appear if they arise as a consequence of density dependent predation.
The experiment might be of limited value if mussels are not the exclusive or most common food item in the diet of the crabs (as it is suggested in lines 358-361).

Validity of the findings

Lines 257-262: You must give a justification for the Kruskal-Wallis and the Welch tests. There are more efficient alternative to those tests.

It is not clear to me how handling times were obtained. Were they estimated or directly calculated?
Main results are a bit disappointing: (i) increasing prey density results in a decreased proportion of animals killed (expected if handling time is relevant); (ii) the proportion killed increases with claw size (expected if claw size influences handling time); and (iii) in one region mortality is larger than in other regions, but this cannot be related to the difference native-invaded range (due to the pseudoreplicated design).

Additional comments

No comment

---

## Round 0.2 · Minor Revisions

The second version of your manuscript has now been reviewed by the same reviewers. Although Reviewer #1 is satisfied with the changes you have made, Reviewer #2 still questions the way you have analyzed your dataset, specifically when dealing with the distribution of the single native and the three invasive populations. Besides other minor remarks, I concur with him/her that you are not using the appropriate statistical tests for conducting a comparison of the distributional ranges of the native vs. invasive ranges, even if this is not the main objective of your work. Either you restrict your analysis to the geographical variability or you will need to use alternative statistical tools if you insist on touching the distributional range topic. Therefore, I am asking you to modify your manuscript accordingly before it can be accepted for publication in PeerJ.

Reviewer 1 ·

Basic reporting

This second version of the manuscript has been modified taking into account my main comments and suggestions included in my previous, full, revision.

I therefore consider that it is now ready to be approved for publication.

Experimental design

This second version of the manuscript has been modified taking into account my main comments and suggestions included in my previous, full, revision.

I therefore consider that it is now ready to be approved for publication.

Validity of the findings

This second version of the manuscript has been modified taking into account my main comments and suggestions included in my previous, full, revision.

I therefore consider that it is now ready to be approved for publication.

Additional comments

This second version of the manuscript has been modified taking into account my main comments and suggestions included in my previous, full, revision.

I therefore consider that it is now ready to be approved for publication.

Reviewer 2 ·

Basic reporting

No comment

Experimental design

Dear Editor/Authors
I still see problems with confounding. Despite the claim of the authors that they do not try to compare native and invasive ranges, that comparison is still implicit in the paper. At several places authors talk about invasive ranges when referring to North American and South African populations and native range when referring to Northern Ireland population. Just to mention three of them: lines 263-264; line 290; lines 338-339.
The authors are right in their rebuttal than native range is not replicated. There is only one range. But a very large one. The native distributional range spans from northern Scandinavia (within the Polar Circle) to northern Mauritania (Fig. 1 in the ms). I have serious doubts that the Northern Ireland population is representative of the whole native range. Any comparison involving native and invasive ranges should consider replicated native regions. Taking more localities in N. Ireland does not solve the problem. Authors say in their response that the N. Ireland population is the baseline population. They are implicitly assuming (i) that populations all along the native distributional range show the same consumer-resource dynamics or, (ii) that the source of North American and South African invaders is the N. Ireland population. Obviously, no evidence for any of these alternatives is presented.
But even in the case that the native range was very limited (for example, restricted to N. Ireland), there are sampling designs and analyses of data which allow for that formal comparison. Asymmetrical designs are commonly used when one of the treatments is un-replicated. Careful thinking is required to enunciate the sequence of tests to detect differences but formal, un-confounded comparisons are possible. Sampling designs used to detect impacts when a single impact locality exists are a good example.
The spatial scale of sampling within regions is also different: probably a few of ten of km for N. Ireland and South Africa and hundreds of km for American regions. No surprise that American localities differ more within regions than the others.

I can recommend publication only if the result of the study is interpreted in terms of geographic variability and not in terms of distributional ranges.

I have two additional comments:
When comparing functional responses (lines 258-268) among regions, inference is done by inspection of overlap of confidence intervals. While this procedure allows for correct inference when intervals do not overlap, when intervals overlap visual inspection might lead to erroneous conclusions. Slope of the initial part of the response curve or asymptote might be formally compared using the bootstrap. Regional differences in functional responses can only be demonstrated if the observed differences among regions are significantly larger than differences among localities within regions. Visual inspection is not enough to conclude. Particularly when localities in some regions exhibit large differences.
I suggest to delete comments on functional responses in the discussion (lines 321 to 331), as a type II functional response might have appeared as an experimental artefact. Authors recognize that in lines 324-328.

Validity of the findings

No comment

Additional comments

No comment

---

## Round 0.3 · accepted · Accept

Dear Brett and co-author,

My name is Xavier Pochon and I was asked to assess your revised manuscript due to the previous editor not being able to finalise this submission. Therefore, I carefully looked through all of the previous reviews and rebuttals, and reviewed the latest version of the manuscript myself. I was very pleased with the way you have addressed the reviewers concerns, and found the revised manuscript to be very interesting and of high overall quality. I have found a few very minor edits that can be easily fixed at the proof stage (see the attached document entitled 'peerj-reviewing-25247-v2_Editor_Annotated.pdf').

Thank you very much for your patience and hard work in improving this study. This will make for a great contribution to the field of marine bioinvasion.

Warm regards,
Xavier

#